Climatic niche comparison between closely related trans-Palearctic species of the genus Orthocephalus (Insecta: Heteroptera: Miridae: Orthotylinae)

Namyatova Anna A. anna.namyatova@zin.ru
1 Laboratory of Phytosanitary Diagnostics and Forecasts, All-Russian Institute of Plant Protection , St Petersburg , Russia
2 Laboratory of Insect Taxonomy, Zoological Institute, Russian Academy of Sciences , St Petersburg , Russia
Bond Jason
Electronic publication date: 2020 Dec 15
Publication date: 2020
Volume: 8
Electronic Location ID: e10517
Received 2020 Apr 29; Accepted 2020 Nov 17
Copyright: ©2020 Namyatova
Copyright year: 2020
Copyright holder: Namyatova
License: This is an open access article distributed under the terms of the Creative Commons Attribution License, which permits unrestricted use, distribution, reproduction and adaptation in any medium and for any purpose provided that it is properly attributed. For attribution, the original author(s), title, publication source (PeerJ) and either DOI or URL of the article must be cited.
License URL: https://creativecommons.org/licenses/by/4.0/

Keywords: Ecological niche modelling, Climate, Distribution, Palearctic, Miridae, Niche overlap, Insects

Funding: Russian Science Foundation grant 19-74-00077 This work was supported by the Russian Science Foundation grant (no 19-74-00077). The funders had no role in study design, data collection and analysis, decision to publish, or preparation of the manuscript.

==============================
Previously climatic niche modelling had been studied for only a few trans-Palearctic species. It is unclear whether and to what extent those niches are different, and which climatic variables influence such a wide distribution. Here, environmental niche modelling is performed based on the Worldclim variables using Maxent for eight species of the genus Orthocephalus (Insecta: Heteroptera: Miridae: Orthotylinae). This group belongs to one of the largest insect families and it is distributed across Palearctic. Orthocephalus bivittatus, O. brevis, O. saltator and O. vittipennis are distributed across Europe and Asia; O. coriaceus, O. fulvipes, O. funestus, O. proserpinae have more limited distribution. Niche comparison using ENMTools was also undertaken to compare the niches of these species, and to test whether the niches of closely related species with trans-Palearctic distributions are more similar to each other, than to other congeners. It has been found that climatic niche models of all trans-Palearctic species under study are similar but are not identical to each other. This has been supported by niche geographic projections, climatic variables contributing to the models and variable ranges. Climatic niche models of all the trans-Palearctic Orthocephalus species are also very similar to two species having more restricted distribution (O. coriaceus, O. funestus). Results of this study suggest that trans-Palearctic distributions can have different geographic ranges and be shaped by different climatic factors.

Introduction

Environmental niches are important characteristics of species. Studying them can help to identify the environmental factors responsible for maximizing the species’ fitness, and lead to a better understanding of how environment is connected to speciation and how closely related species are different in their ecological preferences. Studying climatic niches of widespread species also can help to reveal the climatic variables, connected with the species ability to adapt to different conditions.

The Palearctic spans thousands of kilometers across different biomes and climatic zones. Species occupying large areas of Europe and Asia in this zoogeographic region are called “trans-Palearctic” and are often treated as having the same type of distribution (e.g., Abe, Melika & Stone, 2007; Hubenov, 2008; Potikha, 2015). However, the diversity of such distributions and corresponding climatic niches has never been quantitatively studied.

Most investigations of environmental niche differences treated allopatric vertebrate species (e.g., Losos et al., 2003; Graham et al., 2004; Kozak & Wiens, 2006; Kozak & Wiens, 2010; McCormack, Zellmer & Knowles, 2010; Blair et al., 2013). There are only a few studies of ecological niche similarities of sympatric species (e.g., Knouft et al., 2006; Wellenreuther, Larson & Svensson, 2012; Lisón & Calvo, 2013; Mumladze, 2014; López-Alvarez et al., 2015; Dellicour et al., 2017), and even fewer on insects (Wellenreuther, Larson & Svensson, 2012; Dellicour et al., 2017). Many studies show that closely related sympatric species are different in their ecological niches (Wellenreuther, Larson & Svensson, 2012; Mumladze, 2014; Aguierre-Gutiérrez et al., 2015; López-Alvarez et al., 2015; Dellicour et al., 2017). However, only a few such works treat Palearctic insects (e.g., Wellenreuther, Larson & Svensson, 2012; Dellicour et al., 2017; Avtaeva et al., 2019), and such works on the trans-Palearctic insects are very rare (e.g., Avtaeva et al., 2019). Knowledge on climatic niches is also a prerequisite for studying the influence of climate on speciation, i.e., testing niche conservatism vs niche diversification hypotheses (e.g., Losos et al., 2003; Kozak & Wiens, 2006). Therefore, it is essential to study climatic niches of the species widespread in Palearctic to understand the factors connected with such a wide distribution and how those factors relate to the phylogenetic history.

This current project describes the climatic niches and reveals their differences for closely related species of the Palearctic genus Orthocephalus Fieber, 1858 (Insecta: Heteroptera: Miridae: Orthotylinae). These species inhabit meadows and dry open areas, utilizing numerous species of Asteraceae, where at least the widely distributed taxa are likely to be polyphagous. Orthocephalus has been revised (Namyatova & Konstantinov, 2009), and a morphology-based phylogeny supporting its monophyly has been provided. Currently Orthocephalus includes 23 species, with O. bivittatus Fieber, 1864, O. brevis (Panzer, 1798), O. saltator (Hahn, 1835), and O. vittipennis (Herrich-Shaeffer, 1835) widely distributed in Europe and Asia (Namyatova & Konstantinov, 2009). This allows us to test whether the closely related species with wide distribution in the Palearctic occupy the same climatic niche, or whether their niches are significantly different.

Most of the Orthocephalus species have a low number (<15) of records, except for O. coriaceus (Fabricius, 1777), O. fulvipes Reuter, 1904, O. funestus Jakovlev, 1881, and O. proserpinae (Mulsant & Rey, 1852). The records of these species have also been used to build the climatic niches to find the factors influencing their distribution and compare them to those of the widely distributed species. Those niches will also allow testing whether the climatic niches of the species with trans-Palearctic distribution are more similar with each other than with those of the species having limited distribution. Comparisons of climatic niches of the Orthocephalus species will be the first step in answering the question, whether the niche conservatism or niche divergence or both can relate to speciation in this genus.

The aims of the current work are to: (1) compare the niches of widely distributed species with each other and with those of species with limited distribution to determine the main climatic variables responsible for the trans-Palearctic distribution and limited distribution; (2) to test whether the trans-Palearctic species are significantly similar and whether they are more similar to each other or to the species with a limited distribution; and (3) to draw conclusions on the presence of the niche conservatism or niche divergence processes in speciation within Orthocephalus, based on the phylogeny provided in Namyatova & Konstantinov (2009).

Materials & Methods

Specimens and localities sources

Eight species (and number of unique records) have been analyzed in this work Orthocephalus bivittatus (171), O. brevis (146), O. coriaceus (39), O. fulvipes (18), O. funestus (90), O. proserpinae (19), O. saltator (237), and O. vittipennis (208). Orthocephalus vittipennis is recorded from Western Europe to eastern Yakutia, including numerous records from Central Asia. The distributions of O. saltator and O. brevis are similar to that of O. vittipennis, but among Central Asian countries, they are known only from Kazakhstan. Orthocephalus bivittatus is not recoded from the northern Europe and eastern Siberia, however, is common in Central Asia (Kerzhner & Josifov, 1999; Namyatova & Konstantinov, 2009). Orthocephalus coriaceus is mostly known from the middle and northern Europe with few specimens recorded from Kyrgyzstan. Orthocephalus funestus is known only from Northeast Asia. Orthocephalus proserpinae and O. fulvipes inhabit the Mediterranean region, O. fulvipes is additionally known from Arabian Peninsula and Iran (Kerzhner & Josifov, 1999; Namyatova & Konstantinov, 2009). Numerous collecting expeditions have been taken in Central Asia, Mongolia, Siberia, Russian Far East, European part of Russia by the Russian and Soviet entomologists, and their collections are mainly preserved at the Zoological Institute of the Russian Academy of Sciences (Konstantinov & Namyatova, 2019). Although many records from European countries were excluded (see below), this region is still well represented in the current analysis. Therefore, it is likely, that the known distribution for Orthocephalus species probably reflects the real distribution.

Specimens used for this study are mostly preserved at the Zoological Institution of the Russian Academy of Sciences, St Petersburg, Russia. This collection holds one of the largest Palearctic collections of Heteroptera. The most label data for the Orthocephalus specimens are recorded in the Arthropod Easy Capture database (https://research.amnh.org/pbi/locality/). Some specimens are preserved at the Canadian National Collection of Insects, Agriculture and Agri-Food Canada, Ottawa, Canada (CNC), Linnavuori Collection (LCRT), Matocq collection (MATOCQ), Bavarian State Museum in Zoology, Munich, Germany (ZSM), National Museum of Natural History, Paris, France (NMNH), Ribes Collection (JR), American Museum of Natural History, New York, USA (AMNH), Natural History Museum, Geneva, Switzerland (MHNG), Zoological Museum Amsterdam, Netherlands (ZMAN), Carapezza collection (AC), Finnish Museum of Natural History, Helsinki, Finland (MZH), Institute for Biological Problems of Cryolithozone, Yakutsk, Russia (YIB).The specimens have been identified based on the revision of the genus Orthocephalus (Namyatova & Konstantinov, 2009). This revision corrected species identification errors of previous keys (e.g., Kerzhner & Jachewski, 1964; Wagner & Weber, 1965; Wagner, 1974) which were based on variable coloration and not male genitalia structure which proved crucial for accurate determination of the species. To decrease number of erroneous records, in most cases the localities from other papers have been added only if they cited the above-mentioned revision (Kment & Baňař, 2012; Matocq, Pluot-Sigwalt & Özgen, 2014; Heckmann, Strauss & Rietschel, 2015; Sofronova, 2017; Vinokurov, Golub & Zinovjeva, 2017) or provided the detailed illustrations of genitalia structures (Tamanini, 1981). All records from the literature have been added for O. brevis and O. funestus (Ribes, 1989; Melber, Günther & Rieger, 1991; Dioli, 1993; Gorczyca & Chłond, 2005; Arnold, 2008; Lim et al., 2011; Lim et al., 2012; Lim, Park & Lee, 2013a; Lim et al., 2013b; Nikolaeva, 2011; Frieß, 2006; Frieß, 2014; Roháčová, 2007; Cho et al., 2008; Cho et al., 2011; Kondorosy, 2011; Park et al., 2013; Halimi & Paparisto, 2014; Halimi, Paparisto & Topi, 2014; Shi, Li & Bao, 2016; Vinokurov, Kanyukova & Ostapenko, 2016; Gierlański, 2017; Jung, Kim & Duwal, 2017; Kozminykh & Naumkin, 2017; Taszakowski & Pasińska, 2017), as O. brevis can be easily separated from congeners by its widened antennal segment II and O. funestus inhabits Northeast Asia, and this is the only Orthocephalus species known from Northeast Asia (Namyatova & Konstantinov, 2009). The maps with all records used in this study are provided on Figs. 1 and 2 and the list of those records for all species and specimen information are provided in the Data S1.

Figure 1 Maps of records used in the analysis.

(A) O. bivittatus, (B) O. brevis, (C) O. coriaceus, (D) O. fulvipes. The background maps is taken from https://github.com/nvkelso/natural-earth-quick-start/tree/master/50m_raster/NE1_50M_SR_W. The layer with the country borders is taken from https://github.com/petewarden/openheatmap/tree/master/mapfileprocess/test_data/TM_WORLD_BORDERS-0.3.

Figure 2 Maps of records used in the analysis.

(A) O. funestus, (B) O. proserpinae, (C) O. saltator, (D) O. vittipennis. The background maps is taken from https://github.com/nvkelso/natural-earth-quick-start/tree/master/50m_raster/NE1_50M_SR_W. The layer with the country borders is taken from https://github.com/petewarden/openheatmap/tree/master/mapfileprocess/test_data/TM_WORLD_BORDERS-0.3.

Maps

Layers in 5-arc minute (∼10 km) resolution representing different bioclimatic variables have been downloaded from Worldclim, Version 1.4 (https://www.worldclim.org/version1). In general, the finer resolution leads to more accurate predictions, as the data are averaged within the grid cell. For this study coarser resolution was chosen for two reasons. First, it is a trade-off between the high resolution data across large geographic space and computational efficiency, as there is only a little difference between the models built using different scales especially in broadly distributed species (e.g., Araújo et al., 2005; Seo et al., 2009). Seo et al. (2009) also has shown that with a spatial grid size below 16 ×16 km, there is a good agreement among model area estimated for species of all range sizes. Second, the coordinates for many localities are approximate, and high resolution might also lead to the erroneous interpretations (Graham et al. 2005; Hanberry, 2013). Layers have been trimmed for Palearctic (20°N–90°N, −30°W–180°E) and converted to ASCII format in DIVA-GIS (https://www.diva-gis.org/). Those layers have been uploaded to QGIS 3.10 and converted to vector and used to create “samples with data” files (swd files).

Environmental niche modelling

Maxent software (version 3.4.1) (https://biodiversityinformatics.amnh.org/open_source/maxent/) (Phillips, Anderson & Schapire, 2006) was chosen because it performs well in comparisons with other programs especially for rare species. It works with presence-only data and is considered to produce robust results with sparse, irregularly sampled data and minor location errors, which is applicable to museum data (Elith et al., 2006; Elith et al., 2011; Pearson et al., 2007; Kramer-Schadt et al., 2013). The models have been built using swd files and bioclim layers in ASCII format. For the datasets with >50 localities, bootstrap replicated run type with 25% of localities assigned for the random test percentage were applied. Overall, ten replicates were conducted. For the datasets with <50 localities (O. coriaceus, O. fulvipes and O. proserpinae), crossvalidation with the replicate number corresponding to the locality number was used (Pearson et al., 2007; Shcheglovitova & Anderson, 2013).

The data used in this study are biased towards the easily accessed area, as most of the specimens were collected along major roads and railroads in the area currently corresponding to Russia, as well as Caucasus and Central Asia countries. This can exacerbate over-representation of some regions, which can lead to an inaccurate model. Sampling bias can be addressed by reducing the number of occurrence records in oversampled regions using spatial filtering (Kramer-Schadt et al., 2013); however, it can lead to a situation where the number of occurrences is too few to create a reliable model. Additionally, Maxent automatically discards the redundant records, appearing in the single grid cell (Fourcade et al. 2014; Chiarenza et al., 2019), which is ∼10 km in this study. This is a quite large area, which might accommodate different climate regimes. Assigning the larger grid cell was needed for the filtering, and leading to further loss of information on species climatic preferences. Alternatively, it is possible to manipulate the background data by choosing background data with the same bias as occurrences (Phillips et al., 2009; Elith et al., 2011; Kramer-Schadt et al., 2013). As soon as there are <20 occurrences for two of the analyzed species, the bias file approach was chosen. A previous study showed that using biased background data have increased the performance of the model and should be applicable for cases with small numbers of occurrence points (Kramer-Schadt et al., 2013). A bias file was created as a two-dimensional kernel density estimate, based on the coordinates of the occurrence points, using the kde2d function from the MASS package (Ripley et al., 2020) in R. This approach was applied in previous works (e.g., Filazzola, Sotomayor & Lortie, 2018; Mudereri et al., 2020). Bias files were converted to the raster ASCII format and have been implemented into the biasfile option in Maxent. Ten thousand background points, which is the default Maxent setting, was randomly selected from the area denoted in the bias file. The “cloglog” output was chosen for the visualization and further analysis.

Variable selection

Climatic niche modelling with two sets of variables was conducted. To avoid the model overfitting, it is possible to exclude the highly correlated variables and/or tune the model parameters (Merow, Smith & Silander Jr, 2013). It has been shown that Maxent can perform well with the correlating variables with tuned parameters (Merow, Smith & Silander Jr, 2013; Morales, Fernández & Baca-González, 2017; De Marco Junior & Nobrega, 2018); therefore, the first model type includes all bioclimatic variables (CF model). However, to test whether the model with all variables can be overfitted the highly correlated variables for each species were excluded (CR model) (see below for the details).

Parameter adjustment

There are two modifiable parameters in Maxent, which are feature classes and regularization multiplier. They should be adjusted for each particular case to avoid overfitting and/or over-complexity (Morales, Fernández & Baca-González, 2017). Feature classes correspond to the mathematical transformation of the variables and regularization multiplier (beta multiplier) limits the complexity of the model and generates a less localized prediction; i.e., smooths the model (Phillips & Dudík, 2008; Elith et al., 2011; Merow, Smith & Silander Jr, 2013). Best features and regularization multiplier set for the CF models has been selected using ENMeval package in R (Muscarella et al., 2014a; Muscarella, Kass & Galante, 2014b), using Maxent. The models have been tested against the regularization multipliers ranged from 0.5 to 6 and the default feature classes and their combinations, i.e., L, LQ, LQH, H, LQHP, LQHPT (L = linear, Q = quadratic, H = hinge, P = parameter, T = threshold). The method “block” was chosen, because it accounts for spatial autocorrelation (Muscarella et al., 2014a). This analysis can result in different parameter sets in different runs, so the analysis was done five times for each species. In case if the analyses resulted in different parameter sets, all of them were kept to run the environmental niche modelling in Maxent. All the parameter sets used for the modelling are provided in the Table S1.

For the CR model, the MaxentVariableSelection package in R (Jueterbock et al., 2016; Jueterbock, 2018) was used. It chooses the best set of variables, which has the lowest AICc value, based on the regularization multiplier and features. Comparisons were performed for the same parameters, as in the case of the ENMeval. A separate run was conducted to test the regularization multipliers for each feature class or combinations of classes. For the background data ten thousand background points were extracted from the bias file raster using R. For each feature (or combination of the features), its best regularization multiplier and variable set was kept for further analysis in Maxent, and they are provided in the Table S1.

Maxent provides the list of the percent contribution (PC) and permutation importance (PI) for each variable in the model. The variables with PC and/or PI values higher than 10% are provided in Table 1 for CF and CR models for each species.

Table 1 Variables, contributing to the models CF (first column for each species) and CR model (second column for each species).

The variables used for modelling are marked with “X”. PC and PI denote the variables having PC and PI higher than 10%. Total area of the suitable conditions projected area for each model is provided in the last raw.

	bivittatus	brevis	coriaceus	fulvipes	funestus	proserpinae	saltator	vittipennis	
	CF	CR	CF	CR	CF	CR	CF	CR	CF	CR	CF	CR	CF	CR	CF	CR	
Bio1 Annual Mean Temp	PC PI	PC PI	PC PI	PC PI	X		X		PI		X		PI		PC PI	X	
Bio2 Mean Diurnal Range	PI		X		X	X	X		X	X	X		X	X	X	PC	
Bio3 Isothermality	PI	X	X		PC PI	PC PI	X		PC PI	PC PI	X		X	X	X	X	
Bio4 Temp Seasonality	PI	X	X		X		X		X		PI	PC PI	X	X	X		
Bio5 Max Temp of Warmest Month	X		X	X	X		X	X	X		X		X	X	X		
Bio6 Min Temp of Coldest Month	X		X		X		X	PI	X	PC PI	X		PI	PC PI	X		
Bio7 Temp Annual Range	PI		X		X		X		X		PC PI	X	PI		X	PC PI	
Bio8 Mean Temp of Wettest Quarter	X	X	X		X		X		X		X		X	X	X	X	
Bio9 Mean Temp of Driest Quarter	X	PC	X	X	X		PC PI		X		PC PI		X	X	PC	PI	
Bio10 Mean Temp of Warmest Quarter	X	PC	X		X		X		X	PC	X		X		X	X	
Bio11 Mean Temp of Coldest Quarter	PC		X		PI		X		X		X		PC		PI		
Bio12 Annual Precipitation	X		X	X	X		X		X		X		X	X	X		
Bio13 Precipitation of Wettest Month	X	X	X	X	X		X		X		X		X	X	X		
Bio14 Precipitation of Driest Month	X	X	PC PI	PC PI	PC	PC	X		X		PC	PC	X	X	X	X	
Bio15 Precipitation Seasonality	PC	PC	X	PI	PI		X		X	PC PI	X	X	X	X	X	X	
Bio16 Precipitation of Wettest Quarter	X		X		X	PI	X	PC PI	PI		X		X		X	PC	
Bio17 Precipitation of Driest Quarter	X		X		X		X		X	PC PI	X		X		X		
Bio18 Precipitation of Warmest Quarter	X	PC PI	X	X	X	X	PC PI	PC PI	PC		X		X	X	X	PC PI	
Bio19 Precipitation of Coldest Quarter	X	X	X		PC	PC	PC PI		X		PC	PC PI	X	PC	X	X	
Area (x106 km2)	9.65	8.05	8.78	8.62	8.19	5.08	8.11	5.76	5.41	3.56	1.83	1.73	10.06	10.10	13.67	12.97	

Model evaluation

For model evaluation, training and test AUC values are provided, which is valid for model comparison over the same study area (Bohl, Kass & Anderson, 2019) (see Table S1). The differences between training and test AUC values and omission error rates have been also compared. It has been shown that the model with high differences between AUC values and omission error rate > 0.1 is likely to be overfitted (Bohl, Kass & Anderson, 2019). In the case of each model type (CF and CR), the model with the relatively high AUC values, low differences between training and test AUC and low omission error rates was chosen for the visualization and niche comparisons.

Environmental niche projection area and climatic variable ranges

The obtained environmental niche models were thresholded using the “Maximum training sensitivity plus specificity Cloglog threshold”, as the thresholds maximizing sensitivity and specificity perform well on presence only datasets (Liu, Newell & White, 2016). The total area of the thresholded niche projection was obtained using QGis 3.10. The thresholded maps were used as masks to trim the bioclim layers to obtain the climatic variables ranges for each model. These areas and variable ranges were used to compare the models. The correlation of climatic variables was estimated for each species separately using Pearson’s correlations (PCor), as it is suitable for continuous variables. In this work PCor ≥ 0.9 is considered strong, as it is usually used to discriminate strongly correlated variables (e.g., Jezkova, Olah-Hemmings & Riddle, 2011; Dellicour et al., 2017). I also considered 0.7 ≤ PCor > 0.9 as significant. Tables with Pearson’s correlations for the each species are provided in the Data S2.

Niche overlap

Testing for niche overlap was performed in ENMTools (Warren, Glor & Turelli, 2010). First, the niche overlap was conducted to get the Schoener’s D (D) and Hellinger distance I (I) metrics, these measure similarities between species habitat suitability models. The values of both metrics ranged from 0 (the niches do not overlap) to 1 (the niches are identical). Comparisons were run between all species within each model type, CF and CR, separately. Second, the “Identity test” for each pair of species was conducted. This test randomizes the occurrences for two species, creating the pseudopopulations, and compares the environmental niches for those datasets, creating permuted D and I values. If the D and I values for the actual data are significantly lower than those of the randomized (permuted) data the niches are interpreted to be different (Warren, Glor & Turelli, 2010). The identity test can be performed only for the same set of environmental layers for both compared species; therefore, the identity test was used for the models with all environmental variables. Third, the background test was also performed. It measures the difference between the similarity of two species on one side and the similarity between species and background of another species on the other side. The test should be undertaken for two sides, as it can yield different results for the reversed comparison. If the D and I metrics for actual niche overlap and obtained with background test are similar, this means that the similarity of niches between two species is the same as expected from random data. If the D and I metrics of actual datasets are higher or lower than those from background test, this means that the niches are more similar or more different than expected from random data respectively (Warren, Glor & Turelli, 2010). The background for each species equals to its bias file.

Maps visualization

All maps have been prepared in QGis 10. The background for the Figs. 1–2 is the layer freely accessible at https://github.com/nvkelso/natural-earth-quick-start/tree/master/50m_raster/NE1_50M_SR_W, and it is not copyrighted. The maps for Figs. 3–6 have been created by the uploading the averaged maps resulted from the Maxent analysis to QGis 10. The country borders layer is freely accessible at https://github.com/petewarden/openheatmap/tree/master/mapfileprocess/test_data/TM_WORLD_BORDERS-0.3, and it is not copyrighted.

Figure 3 Geographical projections of the CF models.

(A) O. bivittatus, (B) O. brevis, (C) O. coriaceus, (D) O. fulvipes. Threshold is indicated with the black line. Colors correspond to the suitability score at the bottom of the figure, with 0 corresponding to the most unsuitable places and 1 corresponding to the most suitable places. The layer with the country borders is taken from https://github.com/petewarden/openheatmap/tree/master/mapfileprocess/test_data/TM_WORLD_BORDERS-0.3.

Results

Model evaluation

All the Maxent models have high discriminative power for the training datasets with high AUC. It is higher than 0.9 in all cases except for O. vittipennis, where AUC ranges vary from 0.87 to 0.89. The models are also able to predict the testing points with very similar AUC values, as in training datasets. The training AUC is higher than the test AUC, and the differences between the models chosen for the comparison vary from 0.002 (CF model for O. bivittatus) to 0.027 (CR model for O. vittipennis). Omission rates for the models chosen for the comparisons vary from 0.0789 (CR model for O. funestus) to 0.1620 (CF model for O. vittipennis). The AUC values and omission rates for all models are provided in the Table S1. The Maxent output files for each model chosen for the visualization and niche comparison are provided in the Data S3. The detailed descriptions of the climatic niches for each species are provided in the Data S4.

Figure 4 Geographical projections of the CR models.

(A) O. bivittatus, (B) O. brevis, (C) O. coriaceus, (D) O. fulvipes. Threshold is indicated with the black line. Colors correspond to the suitability score at the bottom of the figure, with 0 corresponding to the most unsuitable places and 1 corresponding to the most suitable places. The layer with the country borders is taken from https://github.com/petewarden/openheatmap/tree/master/mapfileprocess/test_data/TM_WORLD_BORDERS-0.3.

Figure 5 Geographical projections of the CF models.

(A) O. funestus, (B) O. proserpinae, (C) O. saltator, (D) O. vittipennis. Threshold is indicated with the black line. Colors correspond to the suitability score at the bottom of the figure, with 0 corresponding to the most unsuitable places and 1 corresponding to the most suitable places. The layer with the country borders is taken from https://github.com/petewarden/openheatmap/tree/master/mapfileprocess/test_data/TM_WORLD_BORDERS-0.3.

The model with all variables is supposed to be overfitted because of the correlated variables. Therefore, this model is expected to predict smaller areas of suitable conditions and/or narrower ranges of the climatic variables, than the model with reduced set of variables (see Methods). However, the current results do not support this idea. The thresholded maps of the modelled areas with suitable conditions are provided in Figs. 3–6. Those areas of CF models are larger than or subequal to CR for all the species. In the case of variables ranges no model type is noticeably more restrictive than the other (Figs. 7–11). In rare cases, the CF models show significantly more restricted ranges, rather than CR models (e.g., bio12, bio13 for O. bivittatus and O. proserpinae, bio18 for O. proserpinae). The variables ranges for each model are shown on the Figs. 7–11.

Figure 6 Geographical projections of the CR models.

(A) O. funestus, (B) O. proserpinae, (C) O. saltator, (D) O. vittipennis. Threshold is indicated with the black line. Colors correspond to the suitability score at the bottom of the figure, with 0 corresponding to the most unsuitable places and 1 corresponding to the most suitable places. The layer with the country borders is taken from https://github.com/petewarden/openheatmap/tree/master/mapfileprocess/test_data/TM_WORLD_BORDERS-0.3.

Figure 7 The ranges of (A) bio1,(B) bio2, (C) bio3, and (D) bio4.

For each species, the first (red) line corresponds to the CF model, the second (green) line corresponds to the CR model, and the third (blue) line corresponds to the actual records.

The climatic variables with high PC and PI for each model type and for each species are provided in the Table 1. Both models for the same species have different sets of climatic variables explaining their distribution, and each type of model (CF or CR) has different sets of climatic variables explaining the distribution in comparison between the species.

In most of the models both temperature related and precipitation related variables, are important for the species distribution, except for the CF models for O. saltator and O. vittipennis, having only temperature related variables significantly contributing. Most of the variables appear as important for at least one model, except for bio5 (max temperature of warmest month), bio8 (mean temperature of wettest quarter), bio12 (annual precipitation) and bio13 (precipitation of wettest month). In some cases, the same variable significantly contributes to the both type of models within the same species, i.e., bio1 (annual mean temperature for O. bivittatus and O. brevis), bio3 (isothermality) for O. coriaceus and O. funestus, bio4 (temperature seasonality) for O. proserpinae, bio14 (precipitation of driest month) for O. brevis, bio18 (precipitation of warmest quarter) for O. fulvipes, bio19 (precipitation of coldest quarter) for O. coriaceus and O. proserpinae.

Comparison of the variables for the species with similar environmental niches

Annual mean temperature (bio1) is important for all widely distributed species. In the case of the CR models for O. saltator and O. vittipennis this variable does not explain the distribution much. However, min temperature of coldest month (bio6) is important for this type of model in O. saltator, and mean temperature of driest quarter (bio9) is important for the CR model in O. vittipennis, and bio6 and bio9 significantly correlate with each other and bio1 (PCor > 0.87) in those two species (Table 1). Temperature annual range (bio7) and mean temperature of coldest quarter (bio11) highly contribute to at least one of the models in O. bivittatus, O. saltator and O. vittipennis, and they also significantly correlate with each other in all those species, as well as with bio1, bio6 and bio9 for most of the species. In the models of two species, widely occurring in Central Asia (O. bivittatus and O. vittipennis), mean diurnal range (bio2), bio9, mean temperature of coldest quarter (bio11) and precipitation of warmest quarter (bio18) significantly contribute to at least one of the models. In O. bivittatus and O. brevis precipitation seasonality (bio15) is important for at least one of the models. In O. brevis precipitation of driest month (bio14) significantly contributes to both models. Max temperature of warmest month (bio5), mean temperature of wettest quarter (bio8), annual precipitation (bio12), precipitation of wettest month (bio13), precipitation of driest quarter (bio17) only slightly contribute or do not contribute to the climatic models of the trans-Palearctic species.

Suitable conditions for O. bivittatus are shifted to drier places than in other species, whereas suitable conditions for O. brevis are predicted for the places with higher precipitation than in other species. In contrast to other species, suitable conditions for O. vittipennis are predicted in areas with very low temperatures over the winter and very strong seasonality. The models for O. saltator are similar to O. brevis in precipitation levels and temperature changes around the year; however, suitable conditions of the former are also predicted for the areas with warmer temperatures over the summer, than in O. brevis.

The models of O. coriaceus are more similar than random with all the models of widespread species (Table 2). It is different from all of them in the lower margins for isothermality (bio3) limited with higher values (Fig. 7), and suitable conditions are predicted for the places with low temperatures over summer (bio5, bio10) (Figs. 8 and 9). The models for O. bivittatus and O. vittipennis are additionally different from those of O. coriaceus in variables described in the model descriptions for those species (see Data S4). The models of O. coriaceus are most similar to those of O. brevis and O. saltator, which also occupy almost all Europe. Precipitation of driest month (bio14) significantly contributes to both models for O. brevis and O. coriaceus (Table 1). However, in contrast to O. brevis, suitable conditions for O. coriaceus are modelled for the places with very low values for this variable, as well as for precipitation of the driest quarter (bio17) (Figs. 10 and 11). The models of O. coriaceus differ from O. saltator in the upper margin of the precipitation of the driest month and quarter (bio14, bio17) range limited with higher values (Figs. 10 and 11), and the upper margin of seasonality (bio4) limited with lower values (Fig. 7).

Figure 8 The ranges of (A) bio5, (B) bio6, (C) bio7, and (D) bio8.

For each species, the first (red) line corresponds to the CF model, the second (green) line corresponds to the CR model, and the third (blue) line corresponds to the actual records.

Figure 9 The ranges of (A) bio9, (B) bio10, (C) bio11, and (D) bio12.

For each species, the first (red) line corresponds to the CF model, the second (green) line corresponds to the CR model, and the third (blue) line corresponds to the actual records.

Figure 10 The ranges of (A) bio13, (B) bio14, (C) bio15, and (D) bio16.

For each species, the first (red) line corresponds to the CF model, the second (green) line corresponds to the CR model, and the third (blue) line corresponds to the actual records.

Figure 11 The ranges of (A) bio17, (B) bio18, (C) bio19.

For each species, the first (red) line corresponds to the CF model, the second (green) line corresponds to the CR model, and the third (blue) line corresponds to the actual records.

The models for the northeastern O. funestus are more similar than random with O. brevis, O. saltator and O. vittipennis, and they are most similar to O. brevis and O. vittipennis. The models of O. funestus are different from those of abovementioned three widespread species in isothermality (bio3) range very narrow and shifted towards lower values (Fig. 7). In contrast to all other species, suitable conditions for O. funestus and O. vittipennis are modelled for the places with very low temperatures of coldest month, coldest and driest quarters (bio6, bio9, bio11) and strong seasonality (bio4) (Figs. 7–9). The upper margins of the diurnal range (bio2), isothermality (bio3), temperature seasonality (bio4), temperature annual range (bio7) are limited with the lower values (Figs. 7–8), and the upper margins are limited with higher values for many precipitation variables (bio12–14, 16, 17–19) in O. funestus in comparison with O. vittipennis (Figs. 9–11).

Both, O. funestus and O. brevis have suitable conditions in places with high precipitation over the different seasons (bio12, bio14, bio17) (Figs. 9–11), but ranges of many temperature related variables (bio1, bio5, bio6, bio9, bio11) and seasonality (bio4) are limited with the lower margins in O. funestus than in O. brevis (Figs. 7–9). Orthocephalus funestus differs from O. saltator in the variable ranges modelled for the places with stronger annual temperature changes (bio4, bio7), lower temperatures over the different seasons (bio5, bio6, bio9, bio11) and higher precipitation (bio12–14, 16–19) (Figs. 7–11).

Both O. fulvipes and O. proserpinae inhabit southern areas of European Palearctic (Figs. 1D, 2B), but the variables contributing to their models are different (Table 1). For both species in the CF model the mean temperature of driest quarter (bio9) is important. Precipitation over wettest or coldest quarters (bio16 and bio19) significantly contribute to the models of both species. They also have either precipitation of driest month (bio14) or precipitation of warmest quarter (bio18) with high PC. They have similar ranges for isothermality (bio3) and temperature seasonality (bio4), as well as higher temperatures of driest and coldest periods (boi6, bio9, bio11), and lower precipitations over the driest and warmest periods (bio14, bio17, bio18) (Figs. 7–11). The models of O. fulvipes differ from those of O. proserpinae in many temperature variables limited with higher values (bio5, bio6-bio11) (Figs. 8–9), as well as mean diurnal range and temperature seasonality and precipitation of driest month (bio2, bio4, bio14) (Figs. 7 and 10) .

Niche overlap, identity test and background test

The results for the niche overlap, identity test and background test are shown in the Table 2. The I and D metrics show that the niche overlap between all widely distributed species is relatively high in comparison to cases when widely distributed species is compared with locally distributed species, or locally distributed species are compared with each other (I > 0.8, D > 0.5). Similar values are for the overlap between the following pairs: O. brevis and O. coriaceus, O. saltator and O. coriaceus, O. funestus and O. vittipennis. The largest niche overlap is between O. brevis and O. saltator (I > 0.9 , D > 0.6), as well as between O. brevis and O. vittipennis (I > 0.8 , D > 0.6). However, identity test shows that those values do not reach 5% threshold for permuted I and D values, which means that we cannot conclude that the niches are identical.

The background test shows that all pairs of the widespread species are more similar to each other than expected for both CF and CR models and for comparisons in both sides, based on I and D metrics. The same result is shown for the comparisons of O. coriaceus with all widespread species, as well as for the following pairs: O. funestus and O. brevis, O. saltator and O. vittipennis, O. funestus and O. saltator, O. funestus and O. vittipennis, O. coriaceus and O. proserpinae, O. fulvipes with O. proserpinae. Background test undoubtedly shows that the niches are more different from each other only for O. funestus vs O. proserpinae comparison. In all other cases the results are dubious and differ depending on the type of model and statistical metric. The results also can differ for the pair of species, depending which species is used for the background. Generally, the CR models show more overlap with each other than the CF models, and background test is more often shows that the species are more similar to each other for the CR models. If two widespread species are compared, the background test results in very similar metrics values for both directions. If two species with very different areas of suitable conditions sizes are compared, the results depend on which of them is used for the background. In case when the species with larger distribution area is used as a background, the resulted metrics are lower, and therefore, the analysis shows that two species are more similar to each other than expected more often, rather than in the reversed comparison.

Table 2 Niche overlap (in bold), identity test and background test results.

If the niche overlap values are significantly lower than 5% threshold for permuted values, this means that the models are different. If the niche overlap values are higher or lower than those from background test, this means that the niches are more similar or more different than expected from random data respectively.

	Niche overlap I (CF models)	Niche overlap I (CR models)	Identity test 5% threshold for permuted I values	Background test I values, forward comparison	Background test I values, reverse comparison	Niche overlap D (CF models)	Niche overlap D (CR models)	Identity test 5% threshold for permuted D values	Background test D values, forward comparison	Background test D values, reverse comparison	
bivittatus vs brevis	0.802	0.815	0.96	0.623	0.615	0.549	0.555	0.824	0.330	0.336	
bivittatus vs coriaceus	0.7	0.628	0.928	0.626	0.536	0.39	0.326	0.735	0.334	0.257	
bivittatus vs fulvipes	0.42	0.614	0.9	0.626	0.397	0.19	0.318	0.66	0.333	0.168	
bivittatus vs funestus	0.504	0.649	0.95	0.634	0.433	0.28	0.373	0.78	0.34	0.204	
bivittatus vs proserpinae	0.324	0.375	0.953	0.619	0.285	0.141	0.174	0.776	0.327	0.106	
bivittatus vs saltator	0.854	0.83	0.968	0.622	0.613	0.593	0.575	0.838	0.329	0.337	
bivittatus vs vittipennis	0.812	0.748	0.97	0.619	0.694	0.56	0.479	0.835	0.327	0.404	
brevis vs coriaceus	0.836	0.798	0.902	0.608	0.540	0.522	0.485	0.69	0.332	0.259	
brevis vs fulvipes	0.271	0.651	0.847	0.597	0.398	0.08	0.366	0.623	0.323	0.168	
brevis vs funestus	0.79	0.814	0.94	0.614	0.439	0.505	0.539	0.78	0.335	0.208	
brevis vs proserpinae	0.376	0.416	0.862	0.603	0.293	0.158	0.186	0.636	0.328	0.11	
brevis vs saltator	0.905	0.929	0.961	0.615	0.618	0.724	0.758	0.818	0.335	0.342	
brevis vs vittipennis	0.912	0.867	0.96	0.612	0.694	0.677	0.618	0.823	0.332	0.404	
coriaceus vs fulvipes	0.397	0.697	0.87	0.522	0.382	0.169	0.378	0.633	0.248	0.161	
coriaceus vs funestus	0.493	0.473	0.906	0.538	0.436	0.236	0.21	0.689	0.258	0.208	
coriaceus vs proserpinae	0.533	0.579	0.863	0.526	0.287	0.276	0.292	0.625	0.251	0.108	
coriaceus vs saltator	0.854	0.833	0.915	0.538	0.621	0.57	0.538	0.723	0.257	0.347	
coriaceus vs vittipennis	0.723	0.723	0.930	0.531	0.699	0.41	0.389	0.748	0.253	0.411	
fulvipes vs funestus	0.105	0.532	0.846	0.394	0.418	0.223	0.267	0.602	0.167	0.195	
fulvipes vs proserpinae	0.76	0.711	0.841	0.397	0.282	0.484	0.267	0.593	0.168	0.104	
fulvipes vs saltator	0.444	0.679	0.875	0.394	0.596	0.22	0.4	0.664	0.167	0.326	
fulvipes vs vittipennis	0.274	0.651	0.882	0.386	0.687	0.09	0.337	0.662	0.161	0.4	
funestus vs proserpinae	0.181	0.231	0.824	0.417	0.292	0.062	0.076	0.564	0.195	0.109	
funestus vs saltator	0.646	0.713	0.951	0.435	0.618	0.366	0.431	0.791	0.206	0.343	
funestus vs vittipennis	0.749	0.828	0.952	0.435	0.692	0.467	0.552	0.793	0.206	0.402	
proserpinae vs saltator	0.493	0.533	0.867	0.296	0.604	0.253	0.261	0.627	0.11	0.331	
proserpinae vs vittipennis	0.320	0.397	0.877	0.296	0.685	0.129	0.173	0.662	0.111	0.397	
saltator vs vittipennis	0.846	0.819	0.97	0.613	0.693	0.599	0.573	0.84	0.338	0.403	

Discussion

Climatic niches of Orthocephalus species comparison

The modelled environmental niches for the widespread species cover noticeably different areas. Areas in which conditions are suitable for O. bivittatus correspond with the other species the least. They are mostly projected on Central Asia and south of European part of Russia. (Figs. 3A, 4A). Orthocephalus brevis and O. saltator are more similar, having the largest area of suitable conditions in Western Palearctic, however, the geographic projections of their environmental niches do not cover the Mediterranean region for O. brevis, whereas those areas are suitable for O. saltator (Figs. 3B, 4C, 5B, 6C). Suitable conditions for O. vittipennis extend through entire Eurasia, from Europe to Northeast Asia, including Central Asia, (Figs. 5D, 6D), whereas the Mediterranean area is not suitable for this species. Those differences in the areas of suitable conditions are also supported by the differences in the variables with highest contribution to the climatic models (Table 1) and comparisons of climatic variable ranges (see ‘Results’ for the details). Overall, all analyzed Orthocephalus species have different set of variables, most important for their climatic models and none of the pairs have identical climatic niches, which suggests that the climatic niche is species specific even for the closely related taxa. Finally, the differences in the environmental niches are also supported by the identity and background tests. Although in all widespread species niche overlap is high (I > 0.8, D > 0.6) and background test shows that the similarity between them is higher than that of random data, the identity test does not support the hypothesis that they are identical (Table 2). This supports the previous research, showing that the environmental niches in closely related species might be similar, but not identical (e.g., Wellenreuther, Larson & Svensson, 2012; López-Alvarez et al., 2015; Dellicour et al., 2017).

The climatic niche models of trans-Palearctic species are all more similar to each other than expected. The niches of O. coriaceus and O. funestus, distributed in Europe and Northeast Asia respectively, also highly overlap with some trans-Palearctic species. The climatic niche models of O. fulvipes and O. proserpinae, are not very similar with those of trans-Palearctic species.

Schmitt (2007) made an overview of the European types of distribution, and delimited three main types. Species with the center of dispersal in Mediterranean regions are “Mediterranean”, species having extra-Mediterranean center of dispersal belong to “Continental” type, and species with recent alpine or arctic distribution patterns are called “Alpine” or “Arctic”. Testing the center of the distribution for the Orthocephalus species is beyond the scope of the current paper: a phylogeography analysis combined with modelling of distribution during the glacial period is necessary for such a study. However, it is very likely that the trans-Palearctic Orthocephalus species belong at least to two Schmitt’s groupings (2007). Among the analyzed widespread species, suitable conditions in the Mediterranean region are generally not predicted for O. bivittatus, O. brevis and O. vittipennis and they probably belong to the “Continental” group. However, O. saltator has highly suitable climatic conditions in the Mediterranean region and might have its center of dispersal there (Figs. 5C, 6C).

Maps with the records and the areas of suitable conditions suggests that “trans-Palearctic” distribution is a term of convenience and might include many types of distributions connected with the different climatic conditions. Study of more species from different taxonomic groups, including closely related and unrelated species, are necessary to make conclusions on which types of environmental niches are suitable for the trans-Palearctic species and which climatic conditions are connected with such distributions.

Climatic variables important for the Orthocephalus distribution

Annual mean temperature (bio1), temperature annual range (bio7) and mean temperature of coldest quarter (bio11) are important for the models of at least three widely distributed Orthocephalus species. The results also correspond with the results of environment niche modelling of the trans-Palearctic beetle species Pterostichus oblongopunctatus (Fabricius, 1787), which showed that bio1 and bio11 had high contribution to the model of this species (Avtaeva et al. 2019). Modelling of the trans-Palearctic tick Ixodes ricinus (Linnaeus, 1758) showed that min temperature of coldest month (bio6) is among the variables with the highest PC in the climatic model of this species (Porretta et al., 2013) In all trans-Palearctic species bio6 and bio11 highly correlate with bio1 (PCor > 0.9), and this might mean that the distribution of the trans-Palearctic arthropods is limited at least partly by the winter temperatures. Overall, the set of the variables having high contribution to the climate models is unique for each Orthocephalus species, and analysis of more trans-Palearctic species is needed to make conclusions on how type of distribution correlates with the climatic variables.

Orthocephalus brevis, O. coriaceus, P. obongopunctatus and I. ricinus are mostly distributed in the Western Palearctic, and their climatic models have precipitation of driest month (bio14) or precipitation of driest quarter (bio17) with high PC. This reflects the fact that the most suitable conditions for all those species cover middle and northern Europe, but not Mediterranean zones, and therefore prefer the places with relatively high precipitation around the year. This is also supported by the variable ranges of bio14 and bio17 variables for O. brevis and O. coriaceus, which show that these species inhabit places with high precipitation values over the driest period. Similarly to four mentioned species, Orthocephalus saltator is mostly distributed in Western Palearctic, but the areas with the most suitable conditions cover Mediterranean zones. In case of this species, bio14 and bio17 do not contribute much to the climatic model (<10% of PC and PI).

Although O. coriaceus and O. funestus have very different distribution, isothermality (bio3) has high PC and PI for both models in both species. However, according to the temperature ranges the former tends to inhabit places with high isothermality, whereas the latter prefers the areas with low values of this variable. The study with more European and East Asian species and species from Northeast Asia is needed to confirm that isothermality is connected with such distributions.

The precipitation related variables over the different seasons (bio14, bio15, bio18, bio19) are important for models for two species mostly distributed in the Mediterranean regions, and mean temperature of driest quarter (bio9) has high PC and PI in the CF model. This is similar to the Mediterranean species Tomicus destruens (Wollaston, 1865) (Sánchez-García, Galián & Gallego, 2015). It was shown that the variables bio19 and bio9 are also important for the climatic models of different clades and haplotypes of this species. Study of climatic preferences of more insect species from the Mediterranean region is needed to draw conclusions on how the climate is connected with the species distribution in this region.

Comparison of environmental niches between closely related Orthocephalus species in the phylogenetic context

Although, a morphology-based phylogeny for Orthocephalus has been published (Namyatova & Konstantinov, 2009), there are many unresolved clades, and, therefore, cannot be used to adequately analyze potential phylogenetic signal in climatic related tolerances and environmental niches. However, a few conclusions still can be reached. According to this phylogeny, O. bivittatus, O. coriaceus, O. fulvipes, O. funestus, O. saltator and O. vittipennis have very similar vestiture, color and genitalia, and in some cases the species can be identified only from the males. Those species form a clade, which also includes some other species. Orthocephalus brevis and O. proserpinae are very different morphologically, and are not closely related to other species.

Based on these relationships, it can hypothesized that there are at least three processes in this genus related to climatic niches which might be at play. First, there might be phylogenetic conservatism, at least for some climatic variables. For example, O. funestus and O. vittipennis are very similar morphologically. They mostly differ only in hemelytron coloration in males. According to the current analysis, their niches are also more similar to each other than to random data, and both those species can tolerate strong seasonality and very low winter temperatures. Another example is O. fulvipes, inhabiting southwestern Palearctic, which morphologically is very similar to O. saltator, and the latter is the only widespread species in which the Mediterranean region is suitable.

Second, climatic niche convergence is also observed. This study found that climatic niches are very similar in O. brevis and O. saltator, as well as O. brevis and O. vittipennis, and those pairs are not closely related.

Third, the analysis shows the possibility of the distinct niche divergence in the distantly related taxa. According to the background test, the niches are undoubtedly more different from each other than from random data only for the O. funestus and O. proserpinae pair. According to Namyatova & Konstantinov (2009), those two species are not closely related. Those significant niche differences might be explained by differences in habitats, occupied by those two species. Orthocephalus funestus inhabits places in Northeast Asia with wide ranges of temperatures and high precipitation, whereas O. proserpinae lives in the Mediterranean region with narrow temperature ranges and low precipitation (Figs. 5B, 5D, 6B, 6D). However, O. fulvipes, which is more closely related to O. funestus, also prefers dry conditions including Mediterranean and desert climates (Figs. 3D, 4D). The background test for this species pair provides inconsistent results, suggesting that the niches between O. funestus and O. fulvipes are more similar with each other, than those of O. funestus and O. prosepinae. This leads to the hypothesis that the closely related species in Orthocephalus cannot diverge very quickly.

However, the robust molecular-based phylogeny and niche models for other Orthocephalus species are needed to test all those hypotheses on the niche evolution in this genus.

Based on this phylogeny, it is unclear whether the ability for the wide distribution has phylogenetic signal. On one hand, it is very likely that two closely related species can similarly adapt to the climatic conditions (e.g., Losos, 2008; Wiens et al., 2010). On the other hand, even though both sister species can potentially tolerate wide range of climatic conditions, one species might have significantly limited realized niche and distribution because of the strong competition with its sister species.

Conclusions

The study on the climatic niche modelling for eight insect species with trans-Palearctic distribution from the genus Orthocephalus has been performed. The niches of widely distributed trans-Palearctic species (O. bivittatus, O. brevis, O. saltator, O. vittipennis) are very similar to each other, but not identical. The differences are confirmed by the “Identity test” and “Background test” in ENMTools, as well as the comparison of the climatic variables contributing to the modes and variable ranges for the areas, covered by preferable conditions of different species. The niches of the trans-Palearctic species are also similar to two species having more limited distribution (O. coriaceus, O. funestus). Overall, the similarity of the niches of widely distributed species is higher than in cases when the niches of widely distributed and locally distributed species or only locally distributed species are compared. The annual mean temperature significantly contributes (bio1) to the climatic models of all trans-Palearctic species. Other temperature related variables, i.e., min temperature of coldest month (bio6), temperature annual range (bio7), mean temperature of driest quarter (bio9), mean temperature of coldest quarter (bio11) are likely to be important for the climatic niches of the trans-Palearctic species. For the trans-Palearctic species, widely distributed in Central Asia, mean diurnal range (bio2), min temperature of coldest quarter (bio11) and precipitation of warmest quarter (bio18) are also important for at least one of the models. For the Western Palearctic species with the most suitable conditions corresponding to the areas outside the Mediterranean regions (O. brevis and O. coriaceus), precipitation of driest month (bio14) is important. Isothermality (bio3) has high PC and PI for European O. coriaceus and East Asian O. funestus. For the species, distributed in the Mediterranean region (O. fulvipes and O. proserpinae), precipitation related variables over different seasons (bio14, bio16, bio18, bio19), significantly contribute to at least one of the model each. The discussion of the results based on the phylogeny suggests that within Orthocephalus there might be different processes connected to the climate niche differentiation, such as niche conservatism, niche convergence and niche divergence. More studies of climatic niches of the species distributed in the Palearctic are needed to better understand the types of possible climatic niches of widespread species, the main climatic variables shaping the distribution of the Palearctic taxa and how the climatic niches are related to phylogenetic history.

Supplemental Information

Supplemental Information 1 Species records and specimen information.

Click here for additional data file.

Supplemental Information 2 Pearson correlation.

Click here for additional data file.

Supplemental Information 3 Maxent outputs.

Click here for additional data file.

Supplemental Information 4 Climatic niche descriptions for Orthocephalus bivittatus, O. brevis, O. coriaceus, O. fulvipes, O. funestus, O. proserpinae, and O. vittipennis

Click here for additional data file.

Supplemental Information 5 AUC values and omission rates.

Click here for additional data file.

I am grateful to the curator of the Heteroptera collection (Zoological Institution of the Russian Academy of Science), Fedor Konstantinov and the Head of the Laboratory of Insect Taxonomy (Zoological Institution of the Russian Academy of Science), Sergey Sinev, for the access to the collection. I thank Liudmila Osipova (International Council of Clean Transportation, Berlin, Germany) for the assistance with the niche modelling methodology and Elena Pazhenkova (St Petersburg State University, St Petersburg, Russia) for consultations on graphs visualizations with R. I’m grateful to Michael Schwartz (Agriculture & Agri-Food Canada, Canadian National Collection of Insects, Ottawa, Canada), Michael Elias (New South Wales Department of Primary Industries Biosecurity collections, Orange, Australia) and Igor Danilov (Zoological Institution of the Russian Academy of Science) for helping to revise the manuscript.

Additional Information and Declarations

Competing Interests

Author Contributions

Data Availability

The author declares there are no competing interests.

Anna A. Namyatova conceived and designed the experiments, performed the experiments, analyzed the data, prepared figures and/or tables, authored or reviewed drafts of the paper, and approved the final draft.

The following information was supplied regarding data availability:

The records used in the analysis, as well as accession numbers and specimen information, including hosting collections, are available in the Supplemental Files.

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
