# Peer review of "Climatic niche comparison between closely related trans-Palearctic species of the genus Orthocephalus (Insecta: Heteroptera: Miridae: Orthotylinae)"

_PeerJ, doi:10.7717/peerj.10517_

## Round 0.1 · original submission · Major Revisions

Your paper has now been seen by two reviewers. Both reviews were favorable but pointed out a large number of issues and problems that must be addressed before the manuscript would be acceptable for publication. Foremost, there are considerable editorial challenges - numerous run on sentences and inappropriate syntax (one of the annotated reviews notes many misplaced uses of "the," for example). As such, the papers needs considerable editorial work prior to acceptance. I don't mind assisting with minor editorial issues on the final draft but would ask that you carefully edit the manuscript before re-submission.

More specifically, I agree with both reviewers that the introduction contextually needs work. Your hypotheses, framework for the questions addressed, needs to be explicitly stated. I also agree with R2 that some basic background on natural history is warranted.

Note that both authors comment that Table 1 needs work - it's not intuitive - very confusing.

As noted by R2 you need to justify the resolution used, exclusion of points, as well as other methodological issues (R1 as well).

In your revised manuscript, please carefully address each of the comments as well as document how you addressed each criticism or justify why the critique was not valid.

Reviewer 1 ·

Basic reporting

I found a number of long, difficult to understand sentences sometimes completely unclear. I think the ms language needs to be improved.

Figures 7-11 is redundant in a main text. I suggest to add them as a supplementary material

Experimental design

Comparing the species distribution is an interesting subject though the research idea have to be properly derived. In the introduction, I miss the general idea. To what bases the eight species out of 23 Orthocephalus have been selected, and then why the distribution of selected species are compared to each other? Based on previous knowledge of climate niches, what was the expectation? Why the niches of transpalearctic species are expected to be more similar? ...... I suggest to rewrite introduction to develop research idea/hypothesis in a better way.

Methods: Rather comparing everything to each other and get mess, I suggest to define specific targets for comparison. For instance, the second question in introduction is relevant in this context however this question is blurred and lost subsequently. I believe a number of such question/hypothesis can be easily formulate considering the species range size and distribution patterns (including widespread vs local, geographic location), species phylogenetic/life history closeness etc

Validity of the findings

Conclusions are difficult to follow because it is not effectively related to the questions/ideas provided in the introduction. After reading the discussion I missed something basic, most important let alone the conclusion where for instance a sentence "...each of those species demonstrate a different climatic niche" is totally priceless. I suggest first to better formulate the research idea and then develop discussion accordingly.

Additional comments

The manuscript "Climatic niche comparison between closely related trans-Palearctic species from the genus Orthocephalus (Insecta: Heteroptera: Miridae: Orthotylinae)" reports a results of interesting study based on distribution modelling of members of widespread insect genus. However there are a number of shortfalls need to be addressed. I made a comments, suggestions and questions on the ms text directly in pdf form which I believe is rather easy to solve in the revised version.

Annotated reviews are not available for download in order to protect the identity of reviewers who chose to remain anonymous.

·

Basic reporting

Overall, the author provides the background information necessary to know that very few studies have explored climatic niches of species with trans-Palearctic distributions (especially insects) and why starting to fill this knowledge gap would be greatly beneficial for understanding the similarities/differences of the species’ distributions and the differences in climatic factors contributing to the climatic niche models. However, the introduction could use additional information about the general biology of these insects, such as similarities or differences in morphology, physiology, and/or food sources (if known) that could affect niche preferences.

The English is difficult to comprehend at times with confusing run-on sentences (e.g., lines 71-74, 76-78 202-204, etc.) along with distracting and unnecessary commas (e.g., lines 28-30, 36-39, 68-70, 103-105, etc.). This should be improved upon to ensure that readers can understand the text. I have highlighted many of these areas in the annotated pdf for clarification.

Table 1 – this table is not very intuitive. I would recommend synthesizing the material into a more readable table, potentially with one column per species (e.g., the CF model uses all variables so that column is not necessary if you instead denote PC and PI for each model with symbols and separate total areas with a comma). You could also add a description of what each bioclimatic variable is (e.g., in column 1: Bio1 – Annual Mean Temp).

Table 2 is also not intuitive. I would recommend including an additional explanation in the table description about how to interpret what the numbers mean (i.e., how to interpret significance for identity and background tests from these numbers alone). From my experience, I have usually seen niche identity and background test results depicted as graphs (e.g., Figs 1D and 2D in Warren et al. 2010), which seem much easier for readers to understand than a table. If feasible, it could be helpful to include a graph for each test comparison in the Supplemental material.

Overall, the figures are really nice (i.e., easily distinguishable colors, perfect size). However, I recommend putting either country boundaries or lat-long ranges on maps to make it easier to follow when you describe the distributions in Results. As someone who is not very familiar with that geographic area, it was a little difficult to follow at first pass. Also, I would recommend adding a scale for the suitability scores on Figs 3-6 to make it clear what colors correspond to highly suitable versus not suitable.

The Results section Comparison of the variables for the species with similar environmental niches (or at least some elements of it) seems like it should be in the Discussion section.

Other very minor points:
Line 86: change has to have

Lines 259, 286, 330, 356, 381, 401, 534, 537: you reference the wrong figures – Figure 5 references should be Figure 4 and vice versa (also highlighted in the annotated pdf).

Line 340: change to driest

Line 422 and 424: bio1 and bio7 were both labeled annual temperature range

Line 492: change to widespread

Experimental design

The author provides necessary background information for their research question and clearly defines the aims of this study. Additionally, the Materials and Methods section provides a great amount of detail for performing environmental niche modeling and niche overlap analyses. Pointing out the potential pitfalls and addressing them by comparing different models significantly strengthens the results. A researcher interested in environmental niche modeling for their study system could greatly benefit from this. Well done.

I’m still a little confused about why many records from European countries were excluded – could you expand upon why you chose to do this (was it just to decrease the likelihood of records with wrong species identifications?) and how this sampling bias could potentially affect construction of the climatic niches?

Why did you choose 5 arc-minute resolution for the bioclimatic variables (e.g., was there a trade-off between high resolution across a large geographic space and computational efficiency)?

What was the cutoff value for significantly correlated bioclimatic variables? And why/how did you choose this? For example, Jeskova et al. 2011 (http://doi.wiley.com/10.1111/j.1365-2486.2011.02508.x) considered high correlation between variables when the correlation coefficient was ≥ 0.9.

Line 206: The description for the niche identity test is not intuitive (i.e., it’s a run-on sentence and difficult to understand). I would recommend breaking this sentence up into a few, more detailed sentences to highlight the details for this test. Also, why was the identity test only run with the CF model?

Validity of the findings

As pointed out earlier, listing the potential pitfalls of environmental niche modeling and addressing them by comparing different models significantly strengthens the results. Additionally, you have developed a robust methodological framework for others to test this type of hypothesis for other taxa.

You covered background comparisons resulting in niche conservatism for both widespread and limited distributions in the discussion; however, I did not see any reference to the O. funestus vs O. proserpinae background test comparison resulting in niche divergence. I think it would be useful to devote at least a sentence or two to discuss this result and maybe include a potential hypothesis explaining this.

Additional comments

General recommendation – make sure that when you mention information relevant to a figure or table you refer to that specific figure or table. For example, in one paragraph (lines 264-272) there were several mentions of the contributions made by the bioclimatic variables for each model, yet there was no reference to Table 1 for any of these (e.g., could add (see Table 1 for summary) somewhere at the beginning to clarify where this information is coming from).

---

## Round 0.2 · Minor Revisions

Attached is a quick edit through the manuscript using your tracked changes document. The paper still requires some extensive editing but is getting close - please look over my edits (emailed separately).

With respect to content unfortunately you rejected some of the more key, substantive comments by the reviewers that must be addressed, in my opinion. Most notably the following:

1) I asked in my comments that you better focus the introduction on the questions and hypotheses being tested. I agree that exploratory analyses are OK (maybe you need to be explicit about this) but I also think that the more interesting questions are lost in the details. To some extent you are asking questions about niche divergence, niche conservatism, and potential climatic factor related to widespread vs. narrowly distributed taxa – all elements that need to be introduced up front in the manuscript.

2) Reviewer 1 asked that you firm up the conclusions - they stated that the conclusions were difficult to follow because it was unrelated to the introduction. Unfortunately you dismissed the reviewer’s suggestions. If this reviewer is telling you that the conclusions are difficult to follow then the manuscript is not clearly written and the ideas not well articulated. My comments re the introduction might help in this regard but they need to be tied to the intro - otherwise it seems to be very disconnected.

3) Reviewer 1's comments regarding the information starting on line 249 (original manuscript) were also largely ignored. I wholeheartedly agree with the reviewer. This section of the results is long, uninteresting, and nearly impossible to read. Please shorten these species profiles or shift to an appendix.

4) The manuscript could still use a bit of wordsmith-ing to improve clarity. I did a bit of editing but there still remains a need for additional review - lots of superfluous "the", and tense changes within sentences.

---

## Round 0.3 · accepted · Accept

Thanks for taking the time to address all of my comments and for carefully editing the manuscript. Looks good and is now, in my opinion, ready for publication.